# Surface Characteristics and Microbiological Analysis of a Vat-Photopolymerization Additive-Manufacturing Dental Resin

**DOI:** 10.3390/ma15020425

**Published:** 2022-01-06

**Authors:** Ericles Otávio Santos, Pedro Lima Emmerich Oliveira, Thaís Pereira de Mello, André Luis Souza dos Santos, Carlos Nelson Elias, Sung-Hwan Choi, Amanda Cunha Regal de Castro

**Affiliations:** 1Department of Pediatric Dentistry and Orthodontics, School of Dentistry, Federal University of Rio de Janeiro, Rio de Janeiro 21941617, RJ, Brazil; dr.ericlesotavio@gmail.com; 2Department of Pediatric Dentistry and Orthodontics, School of Dentistry, Escola Superior São Francisco de Assis, Santa Teresa 29650000, ES, Brazil; pedroemmerich@hotmail.com; 3Laboratory for Advanced Studies of Emerging and Resistant Microorganisms, Department of General Microbiology, Institute of Microbiology Paulo de Góes, Federal University of Rio de Janeiro, Rio de Janeiro 21941902, RJ, Brazil; thaispdmello@gmail.com (T.P.d.M.); andre@micro.ufrj.br (A.L.S.d.S.); 4Department of Mechanical Engineering and Materials Science, Military Institute of Engineering, Rio de Janeiro 22290270, RJ, Brazil; carloseliasime@gmail.com; 5Department of Orthodontics, Institute of Craniofacial Deformity, Yonsei University College of Dentistry, Seoul 03772, Korea

**Keywords:** digital technologies, biomaterials, poly methyl methacrylate resin, photo-polymerizable resin, surface properties, antifungal activity, antibacterial activity

## Abstract

The wide application of additive manufacturing in dentistry implies the further investigation into oral micro-organism adhesion and biofilm formation on vat-photopolymerization (VP) dental resins. The surface characteristics and microbiological analysis of a VP dental resin, printed at resolutions of 50 μm (EG-50) and 100 μm (EG-100), were evaluated against an auto-polymerizing acrylic resin (CG). Samples were evaluated using a scanning electron microscope, a scanning white-light interferometer, and analyzed for *Candida albicans (CA)* and *Streptococcus mutans (SM)* biofilm, as well as antifungal and antimicrobial activity. EG-50 and EG-100 exhibited more irregular surfaces and statistically higher mean (Ra) and root-mean-square (rms) roughness (EG-50-Ra: 2.96 ± 0.32 µm; rms: 4.05 ± 0.43 µm/EG-100-Ra: 3.76 ± 0.58 µm; rms: 4.79 ± 0.74 µm) compared to the CG (Ra: 0.52 ± 0.36 µm; rms: 0.84 ± 0.54 µm) (*p* < 0.05). The biomass and extracellular matrix production by *CA* and *SM* and the metabolic activity of *SM* were significantly decreased in EG-50 and EG-100 compared to CG (*p* < 0.05). *CA* and *SM* growth was inhibited by the pure unpolymerized VP resin (48 h). EG-50 and EG-100 recorded a greater irregularity, higher surface roughness, and decreased *CA* and *SM* biofilm formation over the CG.

## 1. Introduction

The oral cavity is constituted of a complex and diverse microbiota that influences oral and systemic health [1], such as *Streptococcus mutans*, a major cariogenic bacterial species [2], and *Candida albicans*, a fungal species able to act as an opportunistic pathogen leading to mucosal and disseminated infections [3] under conditions that trigger the imbalance of the oral and immune system homeostasis [4,5,6]. Thus, the introduction of restorative or prosthetic dental materials into the oral environment might provide such an imbalance, thereby maximizing biofilm formation [7]. Recently, products manufactured with three-dimensional (3D) printing have become popular in dentistry, being increasingly used due to the ease of acquisition, processing, and manipulation of the images obtained through intraoral scanning [8,9].

The additive-manufacturing technologies: vat-photopolymerization, power bed fusion, binder jetting, material extrusion, direct energy deposition, material jetting and sheet lamination enable the application to substrates such as polymers, metals and ceramics, yielding different results regarding accuracy, precision and trueness, thus providing diverse and wide applications [10,11,12]. Digital-light-processing (DLP) technology is one of the first 3D-printing processes, which is categorized according to ISO/ASTM 52900/15(E) as a vat-photopolymerization (VP) process, wherein the photopolymerizable liquid is selectively cured by light projection of the object to be printed through a small projector or mirrors that display a single image at one time [13]. Each layer is photoactivated so that the resolution of the item to be printed is directly related to the number of projectors and mirrors used in the technique. That way, at each activation, solid layers of small cubic blocks, referred to as voxels, originate from the interaction of the light with the resin of origin, until the object is completely formed [10,14,15,16,17].

Dental crowns and prostheses, surgical guides and components, orthodontic aligners and appliances, as well as courseware for research and teaching, represent a reality arising from digital dentistry [10,18,19]. However, certain parameters need to be observed in order to obtain the best results from printing techniques applied in dentistry, such as the thickness of each photoactivated layer, base design, post-processing, and storage media, which are all factors that are known to influence the accuracy of 3D dental models [20]. As for printing resolutions, Shim et al. [21] and Unkovskiy et al. [22] demonstrated, through in vitro studies, that greater accuracy is obtained when objects are printed in a 90° orientation in both stereolithography (SLA) and DLP techniques. Alternatively, Hada in 2020 [23] reported that the impression in a 45° orientation presented superior accuracy in relation to 0° and 90° orientations. Shim et al. [21] also demonstrated, recently, that the printing orientation also influences the microbiological colonization capacity, as significantly lower levels of *C. albicans* were detected on objects printed in a 90° orientation rather than 45° and 0° orientations.

With respect to biocompatibility and activity against potential pathogens, photopolymers have been evaluated [24,25,26] regarding the antibacterial capacity of natural polymers [27] and the addition of antimicrobial agents or drugs in the synthetic polymer matrix [28,29,30]. Bloukh et al. [31] emphasized that the microbial-resistance phenomenon is a serious matter that threatens humanity’s existence, mainly due to the indiscriminate use of antibiotic drugs. In dentistry, the development of alternative antimicrobial properties of surgical sutures [32], dental implants [33,34], restorative and denture materials [35,36] are relevant contributors to the mitigation or cessation of the use of antibiotics [37] in clinical practice.

Although there is an exponential growth in the application of additive manufacturing in dentistry, the knowledge regarding the comparison of VP dental resins to auto-polymerizing acrylic-resin systems, which are standardly applied to the manufacture of dental devices, is limited. Therefore, the present research focused on the investigation of a dental VP resin regarding surface characteristics, microbiological colonization, and antifungal/antimicrobial activities against an auto-polymerizing acrylic-resin system as the control. The null hypothesis of our study comprised: (i) the surface characteristics and biofilm formation by *C. albicans* and *S. mutans* of a VP dental resin for additive manufacturing will be similar to an auto-polymerizing acrylic resin, (ii) the 3D-printing resolution will not affect the surface characteristics and microbiological colonization of VP dental resins, and (iii) *C. albicans* and *S. mutans* growth will not be inhibited by the unpolymerized VP dental resin.

## 2. Materials and Methods

### 2.1. Materials

The sample of this study consisted of 300 specimens (4 mm in diameter × 1 mm in height) (Figure 1), divided into three groups (n = 100): control group (CG), constituted of auto-polymerizing acrylic resin, and experimental groups containing VP additive-manufacturing dental resins printed at resolutions of 50 μm (EG-50) and 100 μm (EG-100) (Table 1).

The CG was manufactured with an auto-polymerizing acrylic-resin system (OrtoClass, São Paulo-SP, Brazil) by the same operator, with the aid of prefabricated molds of addition silicone (3 M^®^, São Paulo, SP, Brazil) according to the manufacturer’s recommendations. To reduce the chances of air-bubble occurrence, the specimens were fabricated by an incremental-build-up technique and immersed in a pan with negative pressure (Protécni, Araraquara, SP, Brazil) during the polymerization phase. Then, the finishing of the specimens was performed sequentially using 400, 600, and 1200 g sandpapers in a Politriz metallographic machine (APL4, Arotec, Cotia, SP, Brazil) at a speed of 300 rpm/60 s for each sandpaper per specimen [38].

In the experimental groups, EG-50 and EG-100, the specimens were designed using open-source CAD software (Meshmixer v. 3.5, Autodesk, Inc., San Rafael, CA, USA) (Figure 2) with a 90° print orientation. 3D specimens were manufactured in a dental resin for additive manufacturing (Cosmos Splint, Yller, Pelotas, RS, Brazil) in a 3D printer (PHOTON S, Anycubic 3D Printing, Shenzhen, Guangdong) using digital-light-processing technology (DLP/LCD) with printing resolutions of 50 and 100 μm. The residual surface monomers on the 3D-printed specimens were cleaned using a wipe and isopropyl alcohol. Postpolymerization was performed in accordance with the manufacturer’s instructions using a UV-light polymerization unit (CureDen; DENTIS). The support structures were removed using low-speed rotary instruments (Beltec, Araraquara, SP, Brazil).

### 2.2. Analysis of Surface Characteristics

For the purpose of surface characterization, samples from each group were coated with gold particles and analyzed using a scanning electron microscope (Quanta FEG 250, FEI, Eindhoven, The Netherlands) at ×250, ×500 and ×1000 magnifications.

Morphology and surface roughness were evaluated in three samples per group, in duplicate, using a non-contact, three-dimensional, scanning white-light interferometer Zygo NewView 7100 (Zygo, Middlefield, CA, USA) using ×20 magnification lenses. The mean roughness (Ra), root mean square (rms), and mean roughness of the 3rd peak and valley (R3z) were used, with an area of 40 μm × 40 μm. The mean roughness (Ra) was quantified from the arithmetic mean of the absolute values and the ordinate distance of the points of the roughness profile in relation to the mean line within the measurement area. The root-mean-square (rms) deviation was defined as the square root of the mean of the ordinates’ squares of the effective profile in relation to the mean line within the measurement path. Finally, the mean roughness of the 3rd peak and valley (R3z) was obtained by the arithmetic mean of the partial roughness values corresponding to each of the five modules.

### 2.3. Microbiological Analysis

Prior to the beginning of microbiological analysis, the specimens were sterilized by ultraviolet (UV) light for 30 min [39] and then allocated to cell-culture microplates with 96 wells each (Kasvi^®^, São José dos Pinhais, Paraná, Brazil) wherein 1 × 10^6^
*Candida albicans* yeasts and 1 × 10^6^ colony-forming units (CFUs) of *Streptococcus mutans* were added. The systems were kept in an oven (Quimis^®^, Diadema, Brazil) at a controlled temperature of 37 °C ± 1 °C for 24, 48, and 72 h, in brain heart infusion (BHI). After each of the different timepoints, the resins were re-allocated into new 96-well plates, and the biomass, extracellular matrix, and metabolic activity were quantified in triplicate.

The biomass formed by different micro-organisms was evaluated according to the method described by Peeters (2008) [40]. The culture media were discarded and the biomass was fixed with 100% methanol (200 µL) for 15 min. The systems were dried at room temperature for 5 min and then a solution containing 0.4% crystal violet (200 µL) was added, and the plates were incubated for 20 min. To remove excess dye, the systems were washed again with PBS. Subsequently, the biomass was decolorized with 30% acetic acid (200 µL) for 5 min. The bleach solution (100 µL) was transferred to another 96-well plate, and the absorbance was measured at 590 nm using a SpectraMax M3 microplate reader (Molecular Devices, San Jose, CA, USA).

The extracellular matrix was quantified according to the method described by Choi in 2015 [41]. The culture medium was discarded and a solution containing 0.1% safranin (200 µL) at room temperature was added to the wells for 5 min. To remove excess dye, the system was washed with PBS. Subsequently, the extracellular matrix was decolorized with 30% acetic acid (200 µL) for 5 min. The bleach solution (100 µL) was transferred to another 96-well plate, and the absorbance was measured at 530 nm using a SpectraMax M3 microplate reader.

The metabolic activity of the microbial biomass formed after each incubation time was evaluated by reducing XTT (2,3-bis(2-methoxy-4-nitro-5-sulfophenyl)-5-[(phenylamino)carbonyl]-2H-tetrazolium hydroxide (XTT; Sigma-Aldrich, Saint Louis, MO, USA). A solution of XTT at a final concentration of 200 µg/mL and 0.04 mM of menadione was added to the wells for 4 h in the absence of light. After this time, the color change was measured using a SpectraMax M3 microplate reader at a wavelength of 492 nm [36].

### 2.4. Antifungal and Antimicrobial Activity Testing 

The antifungal and antimicrobial activities of unpolymerized VP resin were evaluated at concentrations of 100%, 90%, 80%, 70%, 60%, 50%, 40%, 30%, 20%, 10%, and 0%, diluted in BHI medium according to modified versions of the CLSI M27-A3 and CLSI M07-A9 (2012) for yeasts and bacteria, respectively. To each of the wells (96-well plate), 1 × 10^3^ yeasts of *C. albicans* and 1 × 10^4^ CFU of *S. mutans* were added. The following controls were performed: (i) pure BHI medium, (ii) pure unpolymerized VP resin, and (iii) BHI medium with micro-organisms. The plates were visually read after incubation at 37 °C for 48 h [42,43].

### 2.5. Statistical Analysis

The statistical analysis was performed using the Statistical Package for Social Science IBM SPSS software for Windows (version 20.0, SPSS Inc., Chicago, IL, USA). The adherence to the normal curve was assessed with the Shapiro–Wilk test. After verifying the normal distribution, the control- and experimental-group comparisons of surface roughness and microbiological data were performed using the one-way analysis of variance (ANOVA) followed by Tukey’s and Dunnett’s multiple-comparisons tests, respectively. In all of the analyses, the significance level was set at 5%.

## 3. Results

### 3.1. Surface Characteristics

#### 3.1.1. Scanning Electron Microscope (SEM)

SEM images of the auto-polymerizing acrylic and VP resin samples are illustrated in Figure 3. According to the analysis of the images, a flat but grooved surface was observed in the CG, with clear notches resulting from the polishing sandpapers. A distinguishable, corrugated and more irregular surface was noticed in the experimental groups (EG-50 and EG-100), which might be attributed to the photopolymers’ layer-deposition process (Figure 3).

#### 3.1.2. Surface Roughness

The surface-roughness data are presented in Table 2. Intergroup comparisons indicated a statistically significant difference in all of the roughness parameters evaluated (*p* < 0.05). The experimental groups, EG-50 (Ra: 2.96 ± 0.32 µm; rms: 4.05 ± 0.43 µm) and EG-100 (Ra: 3.76 ± 0.58 µm; rms: 4.79 ± 0.74 µm), recorded the highest mean roughness and root mean square, compared to the control group (Ra: 0.52 ± 0.36 µm; rms: 0.84 ± 0.54 µm). As for R3z, a statistically significant difference was observed only in EG-100 compared to CG (EG-100: 58,324.17 ± 4936.48 nm; CG: 31,022.08 ± 16,470.44 nm) (Table 2). Images of the three-dimensional morphology of the samples from each group are shown in Figure 4.

### 3.2. Quantification and Kinetics of Biofilm Formation

The biomass, metabolic-activity, and extracellular-matrix data of the micro-organisms *C. albicans* and *S. mutans* that were evaluated in the different study groups are presented in Figure 5, Figure 6 and Figure 7, respectively.

A significantly higher quantification of the biomass of *C. albicans* was observed in CG in relation to EG-100 at 72 h (*p* < 0.05). The CG samples also presented a statistically greater quantification of the biomass of *S. mutans* compared to the experimental groups (EG-50 and EG-100) at 24 h and 72 h (*p* < 0.05), and to EG-50 at 48 h (*p* < 0.05) (Figure 5).

No significant differences were observed in the metabolic activity of *C. albicans* among the study groups. However, a statistically higher metabolic activity of *S. mutans* was observed in CG compared to EG-50 and EG-100 at 48 h (*p* < 0.05), and in CG compared to EG-50 at 72 h (*p* < 0.05) (Figure 6).

CG presented a higher extracellular matrix production of biofilm formation by *C. albicans* in relation to EG-50 and EG-100 at 24 h (*p* < 0.05) and 48 h (*p* < 0.01). Also, the biofilm formed by *S. mutans* recorded increased records of extracellular matrix production in CG compared to EG-50 and EG-100 at 24 h (*p* < 0.001) and 72 h (*p* ≤ 0.001) (Figure 7).

### 3.3. Antifungal and Antimicrobial Activity Testing

Antifungal and antimicrobial activity testing indicated that, with the exception of pure unpolymerized VP resin (100%), all of the VP-resin dilutions and control systems of BHI medium with micro-organisms presented *C. albicans* and *S. mutans* growth after a 48 h incubation period at 37 °C.

## 4. Discussion

The wide application of additive-manufacturing resources in dentistry implies the further investigation into the capacity of oral micro-organisms to adhere and form biofilm on vat-photopolymerization (VP)-resin substrates. This in vitro study analyzed the surface characteristics of a VP dental resin for additive manufacturing, printed at resolutions of 50 and 100 μm, in order to determine whether there would be a difference in the surface characteristics and microbiological activity compared to an auto-polymerizing acrylic resin. This investigation directly applies to the manufacture of intraoral devices that usually remain in the oral cavity for a long period of time [4,5].

The analysis of the images obtained with SEM (Figure 3) and with the scanning white-light interferometer (Figure 4) reveals the surface grooves caused by the sequential finishing with the sandpaper strips in the CG, and the printing traces of each resin layer deposition in the EG-50 and EG-100.

When investigating the surface quality of a material-jetting additive-manufacturing process, Udroiu et al. [44] observed that the platen orientation and finishing type were influential factors of the surface roughness, of which their best interaction produced roughness values (Ra) of up to 0.5 μm. On the other hand, previous studies reported Ra values ranging from 0.87 to 5.77 µm [45,46,47,48], which were attributed to the methodological difference of the experiments, such as diverse materials, printers, additive-manufacturing technologies, printing parameters, build orientations, and measurement techniques. In the present study, the VP resin samples recorded Ra values that were approximately six to eight times higher (EG-50: 2.9 μm and EG-100: 3.7 μm) compared to the auto-polymerizing acrylic-resin samples (CG: 0.5 μm). Despite the fact that the surface roughness of 3D printed objects has been previously related to the thickness of each layer deposition [49,50], the Ra parameter in the present study was not significantly influenced by the layer thicknesses of 50 and 100 μm.

The literature shows that an increased surface roughness above the Ra threshold of 0.2 μm [51] facilitates a greater microbiological colonization [52,53] owing to features such as the greater area available for micro-organisms adhesion, the protection from shear forces and the chemical changes that favor physicochemical interactions [54]. Conversely, despite the higher average Ra values, EG-50 and EG-100 recorded a decreased *C. albicans* and *S. mutans* biofilm formation in comparison to CG. These findings may be attributed to the influence of the surface energy and the charge of the photopolymers for additive manufacturing on micro-organism colonization and growth [7,55]. Cell-surface charge is essential for modulating microbial adhesion and aggregation [56], and since Steiner et al. [57] presented their discoveries of peptides’ antibacterial activity against both bacteria and eukaryotic cells, these proteins have been employed in polymeric structures. A common feature is the combination of amino acids with polar and non-polar side chains, leading to amphiphilic structures in these peptides [25].

The negative surface charge of *S. mutans* and *C. albicans,* which is attributed to the phospholipids and teichoic acids of gram-positive bacteria and the sialic acid of the *Candida* cell wall [58], is attracted by the positive surface charge of the auto-polymerizing acrylic resin and photopolymers. However, positively, neutral and negatively charged functional groups can vary depending on the polymer’s composition [25,59,60,61].

There is also a strong correlation between attractive hydrophobic and repulsive electrostatic forces with *C. albicans’* ability to adhere to polymeric surfaces, which are important in the initial adherence to the dental resin, providing means for further attachment and colonization [62,63]. The poly methyl methacrylate has monomer units exposed on its surface that interact with the hydrophobic domains, thus favoring the immediate adhesion of the fungi [63,64]. Secondarily to hydrophobic forces, there are electrostatic interactions that favor the adhesion process of micro-organisms [65].

Another factor that may influence microbial colonization is the orientation of the object at the time of photopolymer printing. In 2020, Shim et al. [21] demonstrated that objects printed with a 90° orientation, as well as those printed in the present study, had significantly lower levels of colonization by *C. albicans* than those printed at 0° and 45°. In addition, the 90° printing orientation was chosen as it allowed for less contact with the supporting pillar of the specimen during the printing process, thereby reducing the interference on the sample’s surface.

The potential of microbiological growth inhibition by the photopolymers should also be highlighted. Many authors have previously discussed the benefits of introducing components with antibacterial potential to additive manufacturing, especially against micro-organisms that have, to some degree, bacterial resistance [25,66,67,68]. The use of phytochemical agents has also become a prominent alternative strategy against bacteria and fungi. Moussaoui et al. [69] showed promising results of the antibacterial, antifungal and antioxidant power of *Lepidium sativum* seed oil. Meanwhile, the selection of antimicrobial components needs to be done carefully, as there is a great similarity between human cells and fungi (both eukaryotic organisms), which hampers the use of a substance that only has selective toxicity for pathogens [70].

The antifungal and antimicrobial potential of photopolymers would be crucial in dentistry with regards to the manufacturing of several intraoral devices such as prostheses and orthodontic retainers. The photopolymer evaluated in the present study inhibited the growth of *S. mutans* and *C. albicans* in its pure unpolymerized form for a period of 48 h. However, once diluted, even at the lowest concentration of BHI (90% photopolymer and 10% BHI), there was a positive microbiological growth. Therefore, further investigation is necessary to evaluate whether the inhibition of microbiological growth in the pure unpolymerized photopolymer is due to the lack of BHI medium nutrients or to a soluble component with antifungal and antimicrobial properties that loses its potential of action when in contact with the BHI medium.

## 5. Conclusions

Within the limitation of this in vitro study, the following conclusions can be drawn:VP dental-resin samples recorded a greater irregularity, higher surface roughness, and decreased *C. albicans* and *S. mutans* biofilm formation over the auto-polymerizing acrylic resin.Surface characteristics and microbiological colonization of VP resins were not significantly affected by the 3D-printing resolutions of 50 and 100 μm.VP dental resin inhibited *C. albicans* and *S. mutans* growth only in its pure, unpolymerized form for an incubation period of 48 h.

## Figures and Tables

**Figure 1 materials-15-00425-f001:**
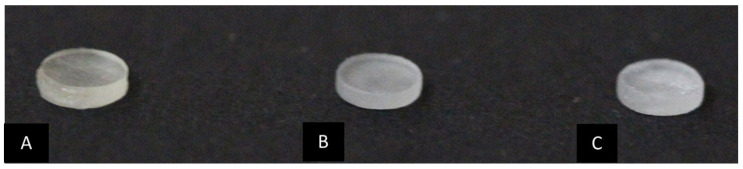
Specimen characteristics. (**A**): CG, auto-polymerizing acrylic resin; (**B**): EG-50, VP additive-manufacturing dental resin printed at 50 μm resolution; (**C**): EG-100, VP additive-manufacturing dental resin printed at 100 μm resolution.

**Figure 2 materials-15-00425-f002:**
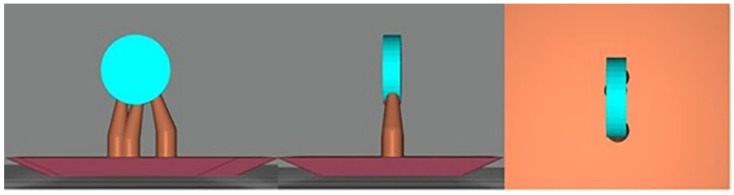
CAD design of the VP additive-manufacturing dental resin sample with 90° printing orientation (Meshmixer v. 3.5, Autodesk, Inc., CA, USA).

**Figure 3 materials-15-00425-f003:**
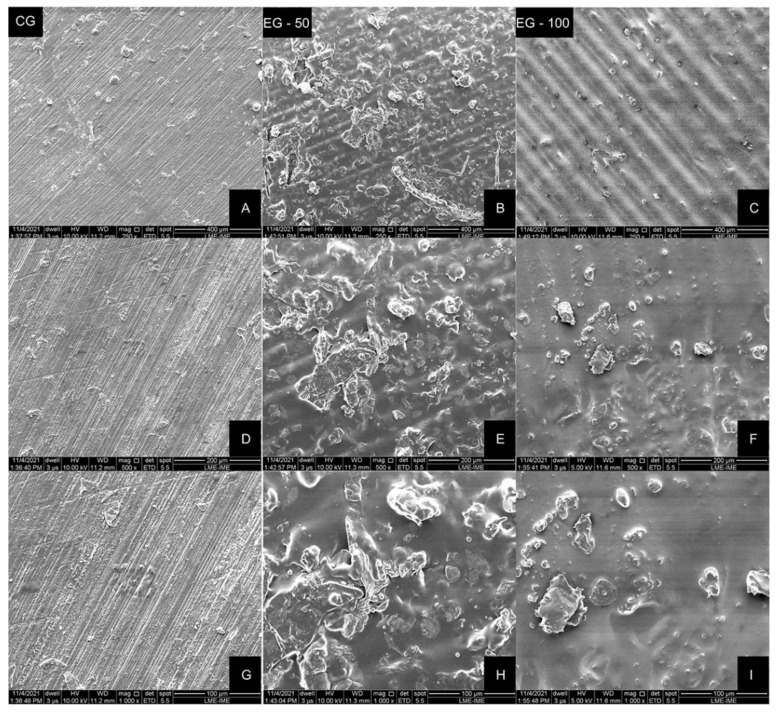
Representative SEM images of specimens from each group. (**A**, **D** and **G**), CG—auto-polymerizing acrylic resin. (**B**, **E** and **H**)—EG-50, VP dental resin printed with 50 μm resolution. (**C**, **F** and **I**)—EG-100, VP dental resin printed with 100 μm resolution. Magnifications: (**A**–**C**), ×250; (**D**–**F**), ×500; (**G**–**I**), ×1000.

**Figure 4 materials-15-00425-f004:**
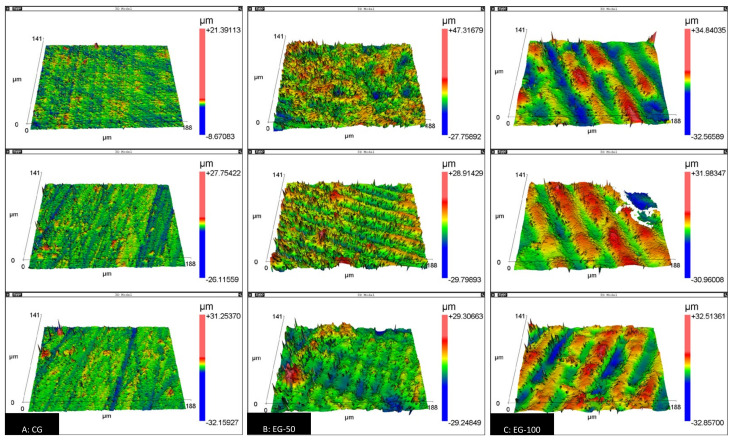
Three-dimensional surface-morphology images obtained with a scanning white-light interferometer (scan area: 40 μm^2^) (**A**): CG, auto-polymerizing acrylic resin; (**B**): EG-50, VP resin printed with 50 μm resolution; (**C**): EG-100, VP resin printed with 100 μm resolution.

**Figure 5 materials-15-00425-f005:**
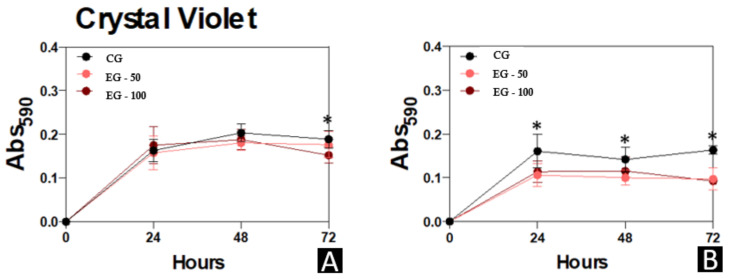
Biofilm formation by *C. albicans* and *S. mutans* for 24, 48, and 72 h at 37 °C. (**A**), *C. albicans*. (**B**), *S. mutans.* At each time point evaluated, the systems were processed to detect biomass by incorporating a crystal-violet solution at 590 nm. Results are expressed as the mean ± standard deviation of three independent replicates. * *p* < 0.05.

**Figure 6 materials-15-00425-f006:**
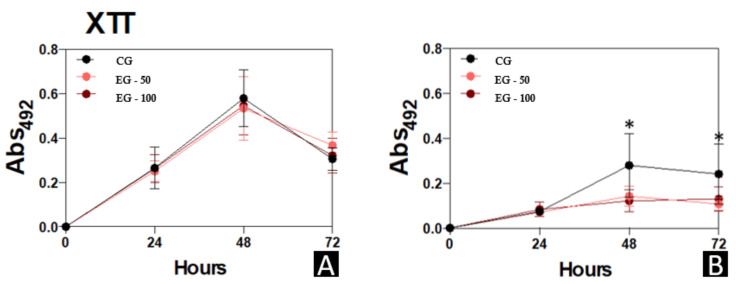
Biofilm formation by *C. albicans* and *S. mutans* for 24, 48, and 72 h at 37 °C. (**A**), *C. albicans*. (**B**), *S. mutans.* At each time point evaluated, the systems were processed to detect metabolic activity by reducing XTT in menadione by viable cells at 492 nm. Results are expressed as the mean ± standard deviation of three independent replicates. * *p* < 0.05.

**Figure 7 materials-15-00425-f007:**
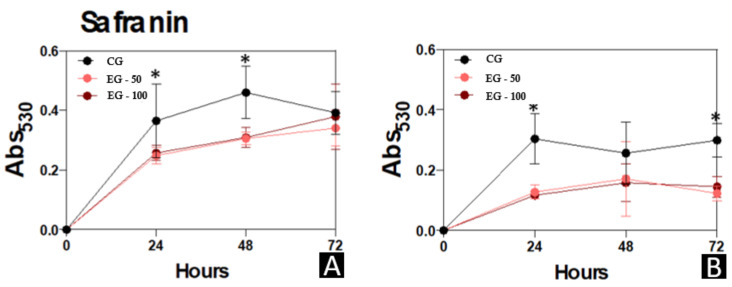
Extracellular matrix in biofilm formed by *C. albicans* and *S. mutans* for 24, 48, and 72 h at 37 °C. (**A**), *C. albicans*. (**B**), *S. mutans. mutans*. At each time point evaluated, the systems were processed to detect extracellular matrix by staining with safranin at 530 nm. Results are expressed as the mean ± standard deviation of three independent replicates. * *p* < 0.05.

**Table 1 materials-15-00425-t001:** Auto-polymerizing acrylic and VP additive-manufacturing resins used in the study.

Product	Component	Manufacturer
Auto-polymerizing acrylic-resin system Ortho Class	Methyl Methacrylate Monomer, DMT, Crosslink, Methyl Ethyl Methacrylate Copolymer	Clássico São Paulo, SP, Brazil
VP additive-manufacturing dental resin COSMOS Splint	Oligomers, Monomers, Photoinitiators, Stabilizer, Pigment	Yller Pelotas, RS, Brazil

**Table 2 materials-15-00425-t002:** Descriptive statistics (mean and standard deviation) of the surface roughness analysis.

Groups	Ra (µm)	*p*-Value	rms (µm)	*p*-Value	R3z (nm)	*p*-Value
CG	0.52 ± 0.36 *a*	<0.001	0.84 ± 0.54 *a*	<0.001	31,022.08 ± 16,470.44 *a*	0.042
EG-50	2.96 ± 0.32 *b*		4.05 ± 0.43 *b*		53,966.22 ± 6866.32 *ab*	
EG-100	3.76 ± 0.58 *b*		4.79 ± 0.74 *b*		58,324.17 ± 4936.48 *b*	

**Ra**, mean roughness; **rms**, root mean square; **R3z**, mean roughness of the 3rd peak and valley. **CG**, acrylic-resin group; **EG-50**, VP resin printed with 50 μm resolution; **EG-100**, VP resin printed with 100 μm resolution. Distinct letters indicate statistically significant difference with ANOVA/Tukey test (α = 0.05).

## Data Availability

The datasets generated and analyzed during the current study are available from the corresponding author upon reasonable request.

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
