# Peer review of "Surface Characteristics and Microbiological Analysis of a Vat-Photopolymerization Additive-Manufacturing Dental Resin"

_materials, 2022, doi:10.3390/ma15020425_

Round 1
Reviewer 1 Report
The authors presented in a simple and direct way the surface characteristics and Microbiological Analysis of a Vat 2 Photopolymerization Additive Manufacturing Resin. The text is interesting to read and opens up new horizons for the reader, taking dentistry to a new level. Digital dentistry will be an easy way to produce excellently fitting restoration by 3D printing. Of course, this new method needs to be thoroughly tested especially related to sustainability and suitability in long term studies. Antimicrobial testing and material durability are major factors in the decision making process.
The authors provided information about the two species C. albicans and S. mutans.
Following points need to be considered:
- in the lines 277-283 provide some basic ideas about the bacterial growth related to the negative surface charge by the phospholipid/teichoic acid of Gram-positive bacteria. C. albicans is not mentioned and shall be added with its outer surface structure properties. This will enrich the explanation of the microbiological testing results. In both cases, the authors may give a deeper insight by mentioning the groups within the VP resin versus the CG, which are interacting with the bacterial and fungal exterior molecular structures.
- What are the relevant reasons/mechanisms by which the surface roughness influences the microbial resistance and biofilm formation ?
- The references need to be changed to the format needed by mdpi
- Additional references can be added related to microbial inhibition on dental material like sutures for the prevention of surgical site infections (see Bloukh et al.) and the microbial resistance phenomenon.
Author Response
Reviewer 1
The authors presented in a simple and direct way the surface characteristics and Microbiological Analysis of a Vat 2 Photopolymerization Additive Manufacturing Resin. The text is interesting to read and opens up new horizons for the reader, taking dentistry to a new level. Digital dentistry will be an easy way to produce excellently fitting restoration by 3D printing. Of course, this new method needs to be thoroughly tested especially related to sustainability and suitability in long term studies. Antimicrobial testing and material durability are major factors in the decision making process.
The authors provided information about the two species C. albicans and S. mutans.
Answer: We are grateful for your comments and appreciation of the manuscript.
Following points need to be considered:
- in the lines 277-283 provide some basic ideas about the bacterial growth related to the negative surface charge by the phospholipid/teichoic acid of Gram-positive bacteria. C. albicans is not mentioned and shall be added with its outer surface structure properties. This will enrich the explanation of the microbiological testing results. In both cases, the authors may give a deeper insight by mentioning the groups within the VP resin versus the CG, which are interacting with the bacterial and fungal exterior molecular structures.
Answer: Thank you for your valuable comment. The Discussion section was improved by adding information about C. albicans and polymers surface charge, as follows: “The negative surface charge of S. mutans and C. albicans, attributed to the phospholipids and teichoic acids of gram-positive bacteria, and sialic acid of the Candida cell wall [58], is attracted by positive surface charge of auto-polymerizing acrylic resin and photopolymers. However, positively, neutral and negatively charged functional groups can vary depending on the polymers composition [25,59-61].”
- What are the relevant reasons/mechanisms by which the surface roughness influences the microbial resistance and biofilm formation ?
Answer: The literature shows that increased surface roughness, above the Ra threshold of 0.2 ?m [51], facilitates a greater microbiological colonization [52,53] owing to features such as greater area available for microorganisms adhesion, protection from shear forces and chemical changes that favor physicochemical interactions [54]. Thank you for your question, this information was added to Discussion.
- The references need to be changed to the format needed by mdpi
Answer: Manuscript was formatted according to the MDPI style. Thank you for your comment.
- Additional references can be added related to microbial inhibition on dental material like sutures for the prevention of surgical site infections (see Bloukh et al.) and the microbial resistance phenomenon.
Answer: Thank you for your valuable contribution. The study from Bloukh et al (2021) and related papers were included in the Introduction, as follows: “With respect to biocompatibility and activity against potential pathogens, photopolymers have been evaluated [24-26] regarding the antibacterial capacity of natural polymers [27] and the addition of antimicrobial agents or drugs in the synthetic polymer matrix [28-30]. Bloukh et al (2021) [31] emphasized that microbial resistance phenomenon is a serious matter that threats the humanity's existence, mainly due to the indiscriminate use of antibiotic drugs. In dentistry, the development of alternative antimicrobial properties of surgical sutures [33], dental implants [34,35], restorative and denture materials [36,37] are relevant contributors to mitigate or discharge the use of antibiotics [32] in clinical practice.”

Reviewer 2 Report
In my opinion, the paper can be published after making some major revisions and some improvements in the presentation of the article, which are as follows:
The abstract does not truly represent the paper. Rephrase the Abstract to become more concise, comprehensible and clear.
The language use in the manuscript is poorly written, in addition to many grammatical and spelling mistakes, poor sentence constructions in many respects, and missing points are left unattended…
The introduction is very poor and less informative. Authors should elaborate their introduction section by citing few more relevant references. The novelty of the work should also be highlighted.
Which are the standards that you used in the synthesis part?
Why you did not use other more powerful techniques in characterization?
The consistency and coherence of the informatons are also missing, and have to be improved.
Comparative analysis of the present data with those published in the literature for the similar type of compounds would support and can improve the quality of discussion.
In part SEM: how the energy of the accelerator beam used?
The quality of SEM figures is too low.
The experimental part must be detailed
Which are the standards that you used in the synthesis part?
Why you did not use other more powerful techniques in characterization?
Kindly update following recent references in the article:
Antibacterial, antifungal and antioxidant activity of total polyphenols of Withania frutescens.L
Bioorganic Chemistry, Volume 93, December 2019, Article 103337
Abdelfattah El Moussaoui, Fatima Zahra Jawhari, Ahmed M. Almehdi,
Author Response
Reviewer 2
In my opinion, the paper can be published after making some major revisions and some improvements in the presentation of the article, which are as follows:
The abstract does not truly represent the paper. Rephrase the Abstract to become more concise, comprehensible and clear.
Answer: Thank you for your suggestion. The abstract was rephrased by adding information about the novelty of the study, specific results description and objective conclusions.
The language use in the manuscript is poorly written, in addition to many grammatical and spelling mistakes, poor sentence constructions in many respects, and missing points are left unattended…
Answer: Thank you for your comment. The manuscript proofreading was re-checked.
The introduction is very poor and less informative. Authors should elaborate their introduction section by citing few more relevant references. The novelty of the work should also be highlighted.
Answer: Thank you for your suggestion. Introduction section was improved by adding additional information and relevant papers.
Which are the standards that you used in the synthesis part?
Answer: A qualitative description of SEM images was provided. Surface roughness was quantitively analyzed with the mean roughness (Ra) in µm, the root mean square (rms) in µm, and the mean roughness of the 3rd peak and valley (R3z) in nm units. The biomass, extracellular matrix, and metabolic activity of biofilm formation was quantified by measuring the absorbance at 590, 530, and 492 nm, respectively. A qualitative analysis of antifungal and antimicrobial activity was performed.
Why did you not use other more powerful techniques in characterization?
Answer: The present paper congregates the first findings obtained from this line of investigation. On the basis of the present study results, further research is ongoing in collaboration with the Department of Microbiology and Department of Materials Engineering. We expect to shortly address the subsequent findings in a forthcoming article format.
The consistency and coherence of the information are also missing, and have to be improved.
Answer: Thank you for your comment. Manuscript was re-checked for consistency and coherence.
Comparative analysis of the present data with those published in the literature for the similar type of compounds would support and can improve the quality of discussion.
Answer: Thank you for your suggestion. Discussion was improved with previous literature in the topic.
In part SEM: how the energy of the accelerator beam used?
Answer: Once the high-energy beam of electrons interact with the sample, secondary electrons, backscattered electrons and X-rays are produced and captured by specific detectors. As the scanned samples were non-conductive, a surface coating with gold particles was provided.
The quality of SEM figures is too low.
Answer: Thank you for your comment. We provided new SEM images at magnifications of x250, x500 and x1000.
The experimental part must be detailed
Answer: Regarding SEM, information about samples surface coating and images magnification is available, as follows: “For the purpose of surface characterization, samples from each group were coated with gold particles and analyzed using a scanning electron microscope (Quanta FEG 250, FEI, Eindhoven, Netherlands) at x 250, x500, and x1000 magnifications.” Please let us know if further specific information should be provided.
Which are the standards that you used in the synthesis part?
Answer: A qualitative description of SEM images was provided. Surface roughness was quantitively analyzed with the mean roughness (Ra) in µm, the root mean square (rms) in µm, and the mean roughness of the 3rd peak and valley (R3z) in nm units. The biomass, extracellular matrix, and metabolic activity of biofilm formation was quantified by measuring the absorbance at 590, 530, and 492 nm, respectively. A qualitative analysis of antifungal and antimicrobial activity was performed.
Why did you not use other more powerful techniques in characterization?
Answer: The present paper congregates the first findings obtained from this line of investigation. On the basis of the present study results, further research is ongoing in collaboration with the Department of Microbiology and Department of Materials Engineering. We expect to shortly address the subsequent findings in a forthcoming article format.
Kindly update following recent references in the article:
Antibacterial, antifungal and antioxidant activity of total polyphenols of Withania frutescens.L
Bioorganic Chemistry, Volume 93, December 2019, Article 103337
Abdelfattah El Moussaoui, Fatima Zahra Jawhari, Ahmed M. Almehdi,
Answer: Thank you for your suggestion. The reference to Moussaoui et al, 2019 was included in Discussion.

Reviewer 3 Report
Thank you for submitting your paper. The work done here draws attention to a significant subject in Photopolymerization of resins. I have found the paper to be interesting. However, several issues need to be addressed properly before the paper is being considered for publication. My comments including major and minor concerns are given below:
- Please consider reviewing the abstract and highlight the novelty, major findings, and conclusions. I suggest reorganizing the abstract, highlighting the novelties introduced. The abstract should contain answers to the following questions:
- What problem was studied and why is it important?
- What methods were used?
- What conclusions can be drawn from the results? (Please provide specific results and not generic ones).
- In the title can the authors indicate that this work is for dental applications.
- The abstract can be improved. Please use numbers or % terms to clearly shows us the results in your experimental work. Please expand the abstract.
- Please consider reporting on studies related to your work from mdpi journals.
- Please add a list of nomenclature before references for all abbreviations, Greek letters, symbols and letters used in the study. (suggested)
- Please consider improving the introduction, provide more in depth critical review about past studies similar to your work, mention what they did and what were their main findings then highlight how does your current study brings new difference to the field.
- The materials and methods section lacks any graphical images which shows test setup, test equipment and some samples, …etc? This is an experimental study and authors should provide sufficient graphical information for the readers to better understand their work and what was done in it.
- Line 186 “which might be attributed to the photopolymers layers deposition process” is this a claim or a fact? In either way the authors can support this sentence with a reference(s) from the open literature if possible.
- Why the authors specifically choose to measure R3z?
- What are the desirable roughness ranges for these resins as per industry recommendations or for optimal performance?
- Is high roughness an issue here? Perhaps even if it is high, it can be easily polished to smooth the outer surface to desirable roughness level or would that be an issue for its application
- In figure 4 and 5 the deviation is high ~ 15-25% which means there is overlapping between different resins. How can the author justify this? Perhaps conducting more experiments to reduce the error bar deviation? Or more robust experimental measurement and analysis? Please justify.
- Some of the results are merely described and is limited to comparing the experimental observation and describing results. The authors are encouraged to include a more detailed results and discussion section and critically discuss the observations from this investigation with existing literature.
- Conclusion can be expanded or perhaps consider using bullet points (1-2 bullet points) from each of the subsections.
Author Response
Reviewer 3
Thank you for submitting your paper. The work done here draws attention to a significant subject in photopolymerization of resins. I have found the paper to be interesting. However, several issues need to be addressed properly before the paper is being considered for publication. My comments including major and minor concerns are given below:
- Please consider reviewing the abstract and highlight the novelty, major findings, and conclusions. I suggest reorganizing the abstract, highlighting the novelties introduced. The abstract should contain answers to the following questions:
- What problem was studied and why is it important?
- What methods were used?
- What conclusions can be drawn from the results? (Please provide specific results and not generic ones).
Answer: Thank you for your suggestion. The abstract was rephrased by adding information about the novelty of the study, specific results description and objective conclusions.
- In the title can the authors indicate that this work is for dental applications.
Answer: Thank you for your suggestion. The term “dental” was included in the title, as follows: “Surface Characteristics and Microbiological Analysis of a Vat Photopolymerization Additive Manufacturing Dental Resin”
- The abstract can be improved. Please use numbers or % terms to clearly shows us the results in your experimental work. Please expand the abstract.
Answer: Thank you for your suggestion. The abstract was improved by specific results description.
- Please consider reporting on studies related to your work from mdpi journals.
Answer: Thank you for your suggestion. Previous studies from MDPI journals were included in the manuscript.
- Please add a list of nomenclature before references for all abbreviations, Greek letters, symbols and letters used in the study. (suggested)
Answer: A list of nomenclature for all abbreviations was provided. Thank you for your suggestion.
- Please consider improving the introduction, provide more in depth critical review about past studies similar to your work, mention what they did and what were their main findings then highlight how does your current study brings new difference to the field.
Answer: Thank you for your comment. Introduction section was improved.
- The materials and methods section lacks any graphical images which shows test setup, test equipment and some samples, …etc? This is an experimental study and authors should provide sufficient graphical information for the readers to better understand their work and what was done in it.
Answer: Thank you for your suggestion. Study samples were illustrated in Figure 1 (Specimens characteristics) and Figure 2 (CAD designs of the VP resin samples with a 90° print orientation).
- Line 186 “which might be attributed to the photopolymers layers deposition process” is this a claim or a fact? In either way the authors can support this sentence with a reference(s) from the open literature if possible.
Answer: Thank you for your suggestion, the sentence was rephrased. “Despite the surface roughness of 3D printed objects have been previously related to the thickness of each layer deposition [49,50], the Ra parameter in the present study was not significantly influenced by the layer thicknesses of 50 and 100 μm.”
- Why the authors specifically choose to measure R3z?
Answer: we included the R3z parameter as it does not consider peaks and valleys that are not representative of the surface. Furthermore, it characterizes very well a surface that maintains a certain periodicity of the grooved profile. Thank you for your comment
- What are the desirable roughness ranges for these resins as per industry recommendations or for optimal performance?
Answer: According to previous literature, desirable surface roughness is up of 0.2 ?m [51] threshold.
- Is high roughness an issue here? Perhaps even if it is high, it can be easily polished to smooth the outer surface to desirable roughness level or would that be an issue for its application
Answer: The literature shows that increased surface roughness, above the Ra threshold of 0.2 ?m [51], facilitates a greater microbiological colonization [52,53] owing to features such as greater area available for microorganisms adhesion, protection from shear forces and chemical changes that favor physicochemical interactions [54].
- In figure 4 and 5 the deviation is high ~ 15-25% which means there is overlapping between different resins. How can the author justify this? Perhaps conducting more experiments to reduce the error bar deviation? Or more robust experimental measurement and analysis? Please justify.
Answer: The high deviation and overlapping between different resins is justified by the sample surface characteristics. Despite the methods standardization, SEM images and surface roughness data shows that samples are slightly different from each other, thus, reflecting on microbiological data.
- Some of the results are merely described and is limited to comparing the experimental observation and describing results. The authors are encouraged to include a more detailed results and discussion section and critically discuss the observations from this investigation with existing literature.
Answer: Thank you for your suggestion. A more detailed discussion of the results was provided.
- Conclusion can be expanded or perhaps consider using bullet points (1-2 bullet points) from each of the subsections.
Answer: Thank you for your comment. Conclusion was expanded in bullet points according to the study hypothesis.

Round 2
Reviewer 2 Report
In the revised version, the authors considerably improved the manuscript according to the recommendations of the reviewers.
Reviewer 3 Report
Please check the paragraphs format (spacing).
The authors provided the answers to the comments from the first round of review and made sufficient changes in the manuscript according to these comments. I recommend this manuscript for a publication in its present form.